# Design and Test of Soil–Fertilizer Collision Mixing and Mulching Device for Manure Deep Application Machine

Yang Niu [1,2], Jiyuan Zhang [1,2], Jiangtao Qi [2,*], Hewei Meng [1,*], Huijie Peng [1,2] and Jiahao Li [1,2]

1    College of Mechanical and Electrical Engineering, Shihezi University, Shihezi 832000, China;
     niuyang@stu.shzu.edu.cn (Y.N.)
2    Northwest Key Laboratory of Agricultural Equipment, Ministry of Agriculture and Rural Affairs,
     Shihezi 832000, China
*    Correspondence: qijiangtao@shuz.edu.cn (J.Q.); menghewei@shuz.edu.cn (H.M.);
     Tel.: +86-1569-9322648 (J.Q.); +86-1336-9935035 (H.M.)

**Abstract:** Aiming at the problems of low uniformity and utilization rate in the traditional deep application method of orchard manure, a soil–fertilizer collision mixing and mulching device was designed. The ditching mechanism, the structure of the soil divider, and the fertilizer delivery auger were analyzed and designed. Furthermore, with the rotational speed of the cutter and auger and the deflection angle of the soil divider as factors, as well as the uniformity of soil–fertilizer mixing and mulching as evaluation indexes, the discrete element simulation tests were conducted. The simulation results showed that when the turret speed, the stirrer speed and the soil separator deflection angle were 140 r/min, 146 r/min and 22°, the mixing uniformity and mulching uniformity were the highest, which was 88.35% and 96.86%, respectively. Based on the optimal parameters, the field test was conducted, and the soil–fertilizer mixing uniformity was 87.02%, with a relative error of 1.33% compared with the simulation test results. The relative error of 94.37% of mulch uniformity is 2.49%, which indicates that the simulation optimization results are reliable and the mixing performance of the device is good and can meet the requirements of soil–fertilizer mixing operation. The results of this study can provide an important reference for the design of the soil and fertilizer mixing machine.

**Keywords:** agricultural machinery; stable manure; mixing; mulching; discrete element simulation

## 1. Introduction

In the process of forest fruit cultivation, manure can improve the soil environment and soil fertility, and its reasonable application can increase the yield and improve the quality of forest fruits [1,2]. In recent years, in order to promote sustainable land development and improve the quality of forest fruit production, relevant state departments have been issuing relevant documents and increasing the proportion of organic fertilizers in agricultural production, such as the "Agricultural Pollution Prevention Policy" formulated by the Ministry of Agriculture, which clearly requires reducing the use of chemical fertilizers and increasing the resource utilization of livestock and poultry manure [3–5].

At present, the manure applied by deep manure is larger, and the manure is not easily absorbed. As a result, most of the existing ditching fertilizer deep application machine mulching does not use soil–fertilizer mixing, making it difficult to play the effect of manure to regulate soil pH and improve the soil environment. With the increasing emphasis on green agriculture, the policy of "organic fertilizer instead of chemical fertilizer" and the agronomic requirements for the application of forest and fruit stall fertilizer are clear, and the application of manure in orchards should be fully mixed with soil before application, which can further improve the efficiency of fertilizer utilization, promote the effective absorption of fruit tree roots, and ensure the effect of fertilizer application and fertilizer utilization efficiency [6,7].

At present, scholars at home and abroad have conducted some related studies on soil and fertilizer mixing. Kwapinska et al. [8] used the discrete unit method to simulate the motion inside a horizontal rotating drum mixer, analyzed the mixing characteristics of the mixture at different rotational speeds, and explored the effects of rotational speed and filling rate on mixing time and mixing number. Ucgul et al. [9] studied a rotating shovel that was able to mix fertilizer from the surface into the soil and improve soil properties. Xiao Hongru et al. [10] designed a 1KS60-35X orchard-type double-spiral trencher and fertilizer spreader, which used double-spiral ditching and simultaneously mixed fertilizer thrown from the middle hollow shaft with soil mixing. The fertilizer application performance was stable in each soil layer and meets the agronomic requirements of orchard fertilization. Huang Yue [11] designed an orchard ditching and fertilizer mixing backfill device, which finished soil–fertilizer mixing in the trench after ditching and applicating and finally mulched the soil. The discrete element model of fertilizer mixing was established by EDEM (particle mechanics simulation software based on discrete element method), and the effect of blades with different curvature radii on the soil–fertilizer mixing uniformity was analyzed. The results showed that it achieved the technical requirement of efficient utilization of soil–fertilizer mixing. Pingping Zhang et al. [12] developed a fruit tree soil–fertilizer mixing precision control fertilizer applicator, where the discharged fertilizer was pushed by a spiral auger into the soil collection box, then mixed with soil evenly, and finally sent by the auger into the ditch to be mulched by the mulching plate. It has high operational efficiency and reduces labor intensity and production cost. Yuan Quanchun et al. [13] designed and studied a soil–fertilizer mixing and layering backfill device. The soil was thrown to the soil–fertilizer mixing and layered backfill device by the ditching device, after which the organic fertilizer was discharged in three ways by the fertilizer discharge mechanism, the backfill fertilizer was conveyed by the auger, and the paddles were spirally arranged between the auger to improve the soil–fertilizer mixing quality, and the soil–fertilizer mixture was backfilled to the fertilizer application trench in turn to realize layered backfill. At present, in the deep application of orchard fertilizer, the soil and fertilizer mixing methods are mainly mixed directly in the ditch and then backfilled to the ditch after being mixed, both of which are mixing soil and fertilizer by stirring, and the operation efficiency is not high.

In this paper, combining agronomy and farmers' needs, a soil–fertilizer collision mixing and mulching device, mixing soil and fertilizer while mulching in ditching operation, was designed. By designing the ditching mechanism, fertilizer discharge mechanism and soil dividing plate, establishing a discrete element simulation model, the best combination of device parameters was established, and field trials were conducted for verification. The hope is that we can realize soil–fertilizer mixing, improve the fertilizer utilization rate, and provide technical support for the fine and deep application of manure.

## 2. Materials and Methods

### 2.1. Overall Structure and Working Principle of Soil Fertilizer Collision Mixing and Mulching Device

#### 2.1.1. Overall Structure

The overall structure of the soil and fertilizer collision mixing and mulching device is shown in Figure 1, which is mainly composed of the ditching mechanism, flow-guided mechanism, soil dividing plate and fertilizer discharging mechanism. The ditching mechanism is mainly composed of a rotary tillage knife and knife plate, and the fertilizer discharging mechanism is mainly composed of auger, a fertilizer box and a scraper.

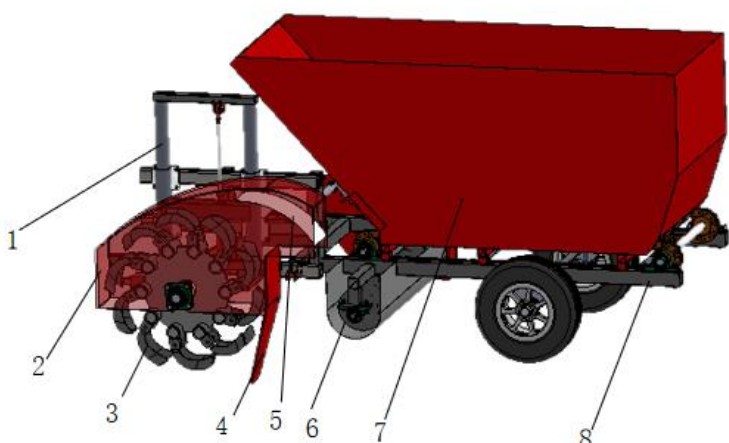

**Figure 1.** General structure of soil and fertilizer collision mixing and mulching device. 1. lifting frame, 2. guide cover, 3. ditching mechanism, 4. trench cleaning plate, 5. Soil dividing plate, 6. Auger, 7. fertilizer box, 8. frame.

### 2.1.2. Working Process

When working, as shown in Figure 2, the ditching mechanism cut soil for ditching with device advancing, and the cut soil moves to the rear under the action of ditching knife and flow-guided mechanism. Then, the back-thrown soil was shunted and delaminated to form soil mixed with fertilizer and soil mulched with fertilizer. The soil mixed with fertilizer, in the back-throwing process, collides and mixes with the fertilizer conveyed by the auger to form soil–fertilizer mixture and then mulch to the trench. Finally, the soil mulched with fertilizer is back-thrown to cover the soil–fertilizer mixture, and the mixing and mulching work in the application of orchard manure is completed.

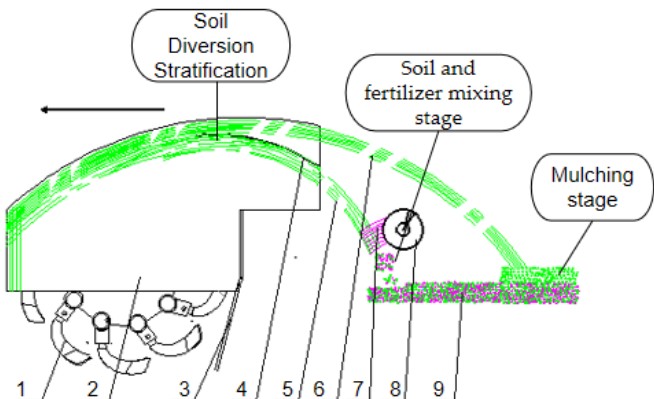

**Figure 2.** Principle diagram of soil and fertilizer collision mixing mulch. 1. Ditching mechanism, 2. Deflector, 3. Ditch cleaning plate, 4. Soil dividing plate, 5. Back-throwing fertilizer mixing soil particles, 6. Back throwing of soil particles covered with fertilizer, 7. Manure particles, 8. Auger, 9. Soil fertilizer mixing layer.

### 2.1.3. Main Technical Parameters

Combined with the agronomic requirements of autumn manure deep application in orchards, and to achieve soil and manure mixing and mulching, the main technical parameters of the manure deep application machine were determined, as shown in Table 1.

**Table 1.** Main technical parameters of the whole machine.

| Parameters | Numerical Value |
|---|---|
| Dimension (L × W × H) (mm) | 3500 × 2000 × 1550 |
| Matching power (kw) | 33~44 |
| Hook-up method | Semi-suspension |
| Operating speed (km/h) | ≥1.2 |
| Fertilization depth (cm) | 0~35 |
| Width of fertilizer application (cm) | 25 |

*2.2. Key Components Structural Parameters Determination*

2.2.1. Ditching Mechanism

The ditching mechanism cuts the soil and completes the furrowing operation while throwing the cut soil back to the inflow mulching component, including ditching the disc, furrowing the knife and driving the mechanism of the cutter disc. The disc-type trencher breaks soil evenly and works efficiently, and it is widely used in various orchards for ditching and fertilization and other agricultural production fields [13]. During the ditching operation, the motion of the ditching blade is synthesized by the horizontal motion of the trencher and the rotational motion of the disc, and the motion trajectory is a coswing [14–16], as shown in Figure 3.

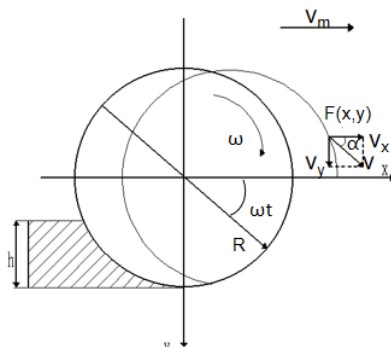

**Figure 3.** Analysis of ditching tool end point motion. *R*—Drenching knife disc end point gyration radius, m; *t*—Time, s; $v_m$—Trencher forward speed, km/h; $\omega$—Knife shaft angular speed, rad/s.

Let the coordinates of any point on the trajectory of the end point of the ditching blade be *F(x, y)*. Take the center of rotation of the ditching blade O as the coordinate origin, the forward direction of the machine is the positive direction of the *x*-axis, and the vertical downward direction of the *y*-axis, as shown in the figure. The equation of the motion trajectory of the point *F(x, y)* is

$$\begin{cases} x = v_m t + R \cos \omega t \\ y = R \sin \omega t \end{cases} \tag{1}$$

Let $\lambda = \frac{v_p}{v_m} = \frac{\omega R}{v_m}$, let $\phi = \omega t$. Substituting into the above equation

$$\begin{cases} x = R(\frac{\phi}{\lambda} + \cos \phi) \\ y = R \sin \phi \end{cases} \tag{2}$$

The tool endpoint at any moment *x*-axis and *y*-axis partial velocity is

$$\begin{cases} v_x = \frac{dx}{dt} = v_m - R\omega \sin \omega t \\ v_y = \frac{dr}{dt} = R\omega \cos \omega t \end{cases} \tag{3}$$

Then, the absolute velocity $v$ at the endpoint of the tool is

$$v = \sqrt{v_x^2 + v_y^2} = \sqrt{v_m^2 + R^2\omega^2 - 2v_m R\omega \sin \omega t} \tag{4}$$

As shown in Equation (4), the higher the rotational speed of the cutter, the greater the linear speed of the cutter edge. However, the increase in cutting speed will also cause the rise in power consumption of soil cutting and throwing. When the cutter speed is low, the soil cannot be thrown out, but it can only be passively pushed upward by the soil below. It may damage the cutter. Combined with the pre-experimental result, the final setting of the cutter speed is 140–160 r/min.

Cutter diameter $D$ is one of its important structural parameters, which has a certain influence on the soil spreading distance of the rotary ditching device, power consumption of the machine and the size of the model, etc. According to the formula

$$D = (1.2 \sim 1.5)H \tag{5}$$

where

$H$—depth of ditching, m.

The ditching depth designed in this paper is 30 cm, which is finally taken as $D = 40$ cm.

### 2.2.2. Fertilizer Delivery Auger

In this paper, a spiral auger was selected to convey manure, and the auger parameters were determined according to the operational requirements [17]. According to the agronomic requirements, the auger fertilizer delivery rate was determined to be 10 t/h, and the auger outer diameter and delivery volume equations are as follows:

$$D \geq K \sqrt[2.5]{\frac{Q}{\varphi \rho c}} \tag{6}$$

$$Q = 47D^2 S\varphi \rho c \tag{7}$$

where

$D$—spiral diameter, m; $Q$—conveying volume per hour, t/h; $K$—material characteristic coefficient; $\varphi$—filling coefficient; $\rho$—material accumulation density, t/m$^3$; $c$—inclination coefficient; $S$—pitch; n—rotational speed, r/min.

The material characteristic coefficient $K$ is 0.05, the filling coefficient is 1, the auger operation is horizontal, so the inclination angle coefficient $C$ is 1, and the screw diameter $D$ is 0.25 m. The pitch $S$ is 0.20 m. According to the auger conveying capacity shown in Formula (7), the minimum winch speed n is 120 r/min to reach the theoretical conveying capacity. The spiral conveying component assembly is shown in Figure 4.

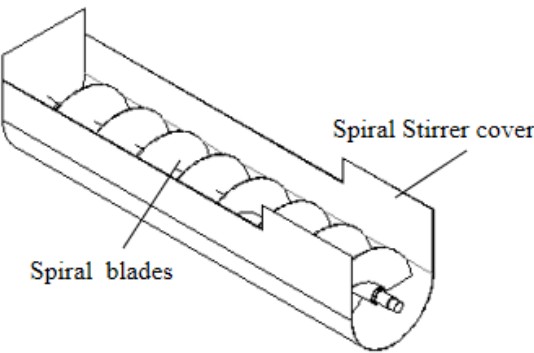

**Figure 4.** Schematic diagram of the structure of screw conveying device.

During the operation of the spiral auger, with the propulsion of the spiral blades, the fertilizer moves along the circumferential direction of the spiral blades while making horizontal movements along the axial direction [18,19]. The velocity analysis of a fertilizer particle on the spiral blade is shown in Figure 5, where the location of the fertilizer particle is at a distance from the axis and the spiral lift angle is $\alpha$. When the spiral blade rotates, the implication velocity of the fertilizer particle is equal to the linear velocity $V_N$ at that point, and the implication velocity direction is perpendicular to the wall spread line of the spiral blade.

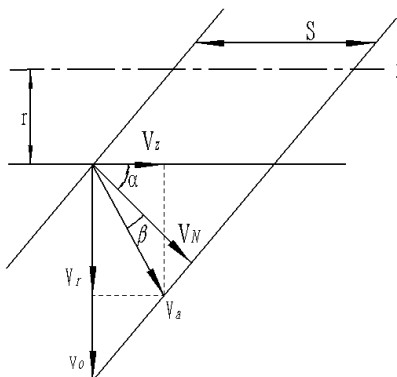

**Figure 5.** Fertilizer speed decomposition diagram. *S*—pitch, mm; *r*—distance between the particles on the spiral surface and the spiral axis, mm; $V_N$—implicated velocity, m/s; $V_a$—absolute velocity, m/s; *Vr*—circumferential directional fractional velocity, m/s; $V_z$—axial directional fractional velocity, m/s; $V_0$—initial velocity, m/s; $\alpha$-spiral lift angle of the spiral blade, °; $\beta$—external friction angle of the fertilizer particles, °.

As can be seen from Figure 5, the absolute velocity $V_a$ of fertilizer in the conveying process can be decomposed into the circumferential directional partial velocity $V_r$ and the axial directional partial velocity $V_z$, whose values are

$$V_z = V_a \cos(\alpha + \beta) \tag{8}$$

$$V_r = V_a \sin(\alpha + \beta) \tag{9}$$

$$V_a = \frac{V_N}{\cos \beta} = \frac{V_0 \sin \alpha}{\cos \beta} \tag{10}$$

$$V_0 = \frac{2\pi n r}{60} \tag{11}$$

$$r = \frac{S}{2\pi \tan \alpha} \tag{12}$$

where: *n*—spiral auger speed, r/min.
According to Equations (10)–(12).

$$V_a = \frac{nS \cos \alpha}{60 \cos \beta} \tag{13}$$

According to Equations (8), (9) and (13) we obtain:

$$V_z = \frac{nS}{60} \left( \cos^2 \alpha - \cos \alpha \sin \alpha \tan \beta \right) \tag{14}$$

$$V_r = \frac{nS}{60} \left( \cos \alpha \sin \alpha + \cos^2 \alpha \tan \beta \right) \tag{15}$$

Furthermore:

$$\cos \alpha = \frac{1}{\sqrt{1 + \tan^2 \alpha}} = \frac{1}{\sqrt{1 + \left( \frac{S}{2\pi r} \right)^2}} \tag{16}$$

$$\sin \alpha = \sqrt{\frac{1}{1 + \frac{1}{\tan^2 \alpha}}} = \frac{S/2\pi r}{\sqrt{1 + \left( \frac{S}{2\pi r} \right)^2}} \tag{17}$$

Coupling (8) to (17).

$$V_z = \frac{nS}{60} \times \frac{1 - \frac{\mu S}{2\pi r}}{1 + \left( \frac{S}{2\pi r} \right)^2} \tag{18}$$

$$V_r = \frac{nS}{60} \times \frac{\mu + \frac{S}{2\pi r}}{1 + \left( \frac{S}{2\pi r} \right)^2} \tag{19}$$

In the working process of soil–fertilizer mixing and mulching device, the speed of the auger shaft directly affects the mixing effect. If the speed is larger, the spiral blade produces larger centrifugal force along the circumference, and the manure particles will be thrown to the edge of the spiral blade. While the material axial speed is faster, the material is quickly discharged. It reduces the mixing effect of soil–fertilizer. If the speed is smaller, the manure will cake due to the increasing squeezing pressure between fertilizer and its high water content. It is not conducive to the mixing. Therefore, based on the theoretical analysis and related research, the auger speed range was set from 130 to 150 r/min according to the pre-experimental result of the device.

### 2.2.3. Soil Separation Plate

The spreading of soil during ditching is often chaotic and irregular, but most soil particle groups will have the same tendency as well as fall within a certain spreading range. So, it is crucial to design a soil-splitting deflector to analyze the landing point of soil particle groups.

The splitting plate not only diverts the back-thrown soil to form the soil mixed with fertilizer and the soil mulched with fertilizer but also controls the amount of both types of soil. Its operation effect directly affects the uniformity of fertilizer mixing and mulching [20]. In this paper, the EDEM simulation software and theoretical calculations were applied to obtain the optimal structural parameters of the soil splitting plate by constructing a model of the soil splitting plate [21,22]. The force analysis of a soil particle on the splitting plate was analyzed as shown in Figure 6.

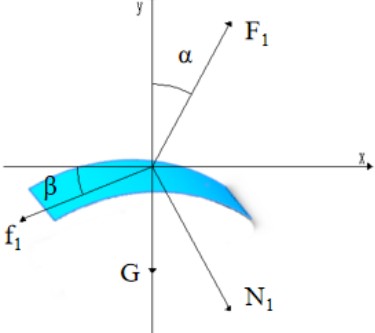

**Figure 6.** Schematic diagram of soil separation plate structure. $N_1$—the force of diversion surface on soil particles, N; $f_1$—the friction force between the diversion surface and soil particles, N; $F_1$—the normal force of trench cutter on soil particles, N; $G$—the gravity of soil particles, N.

The combined external force on the soil particles at point *D* is *F*. Then

$$F = G + N_1 + f_1 + F_1$$

From Newton's second law.

$$F = ma$$

And

$$
\begin{cases}
F_x = F_1 \cos(\frac{\pi}{2} - \alpha) - f_1 \cos\beta - N_1 \cos(\frac{\pi}{2} - \alpha) \\
F_y = F_1 \cos\alpha - f_1 \cos(\frac{\pi}{2} - \beta) - N_1 \cos\alpha
\end{cases}
\tag{20}
$$

The derivation of Equation (20) yields the absolute velocity $V_a$ of the soil particles on the inflow surface, with components in the *x* and *y* axes as

$$
\begin{cases}
v_x = \frac{F_1 \cos(\frac{\pi}{2} - \alpha) - f_1 \cos\beta - N_1 \cos(\frac{\pi}{2} - \alpha)}{m} \\
v_y = \frac{F_1 \cos\alpha - f_1 \cos(\frac{\pi}{2} - \beta) - N_1 \cos\alpha}{m}
\end{cases}
\tag{21}
$$

Different deflection angles of the splitter plates will affect the ratio of the two parts of the soil and the movement of the soil mixed with fertilizer. If the deflection angle of the splitter and the distance between the splitter and the deflector is too small, it will cause the soil to accumulate on top of the splitter and prevent the fertilizer soil from being discharged properly. Based on the above considerations and pre-experiments, the declination angle of the soil divider was finally set at 20~40°.

*2.3. Soil and Fertilizer Collision Mixing Mulching Device Working Process Simulation and Test*

2.3.1. EDEM Simulation Model and Simulation Parameters

The model of the soil–fertilizer collision mixing mulching device was created in SolidWorks software and imported into EDEM software. A soil layer of 3000 mm in length, 600 mm in width and 400 mm in height was created, and the soil color was set to red, while the manure was to yellow for better differentiation. The Hertz-mindlin with the JKR model was used between the soil and the manure, and the Hertz-mindlin (no slip) built-in Optimal model was used between the fertilizer particles and the fertilizer discharge device [23]. According to scaling theory, the simulation with a large particle size did not affect the results after modifying the discrete element model accordingly [24]. Due to the large number of particles, the particle radius was scaled to 5 mm in this paper to shorten the simulation time, and the soil and manure discrete element parameters were referenced as shown in Table 2 [25,26].

According to the simulation pre-experiment, the time step during simulation is $2.5 \times 10^{-5}$ s, the data-saving interval is 0.01 s, and the simulation grid is 20 times the particle radius [27]. The soil–fertilizer collision mixing mulching device moves in the forward direction with a speed of 0.3 m/s, and the operation time is 7 s.

**Table 2.** Discrete element parameters.

| Type | Property | Value |
|---|---|---|
| Soil | density/(g/cm$^3$) | 2500 |
| | Poisson's ratio | 0.25 |
| | Modulus of shear (Pa) | $1 \times 10^8$ |
| | Superficial energy(J/m$^2$) | 8.7 |
| Soil–Soil | Recovery factor | 0.6 |
| | Static coefficient of friction | 0.6 |
| | Rolling coefficient of friction | 0.4 |

**Table 2.** *Cont.*

| Type | Property | Value |
|---|---|---|
| Soil–Steel | Recovery factor | 0.6 |
| | Static coefficient of friction | 0.6 |
| | Rolling coefficient of friction | 0.05 |
| Manure | density/(g/cm$^3$) | 1500 |
| | Poisson's ratio | 0.25 |
| | Modulus of shear (Pa) | $1 \times 10^7$ |
| Manure–Manure | Superficial energy(J/m$^2$) | 0.02 |
| | Recovery factor | 0.6 |
| | Static coefficient of friction | 0.65 |
| | Rolling coefficient of friction | 0.1 |
| Manure–Steel | Recovery factor | 0.6 |
| | Static coefficient of friction | 0.7 |
| | Rolling coefficient of friction | 0.11 |
| Manure–Soil | Superficial energy (J/m$^2$) | 0.0225 |
| | Recovery factor | 0.4 |
| | Static coefficient of friction | 0.66 |
| | Rolling coefficient of friction | 0.18 |
| Steel | density/(g/cm$^3$) | 7850 |
| | Poisson's ratio | 0.3 |
| | Modulus of shear (Pa) | $7.94 \times 10^{10}$ |

### 2.3.2. EDEM Soil Fertilizer Mixing Process Analysis

As shown in Figure 7, during the operation, the ditching and mulching device will produce cutting, throwing and the collision effects on the soil and manure, which affect the soil and fertilizer mixing operation effect. To explain the soil and manure transport mechanism under different stages of the machine operation, the kinetic energy and motion trend analysis of the simulated soil and manure particles are carried out with the EDEM post-processing.

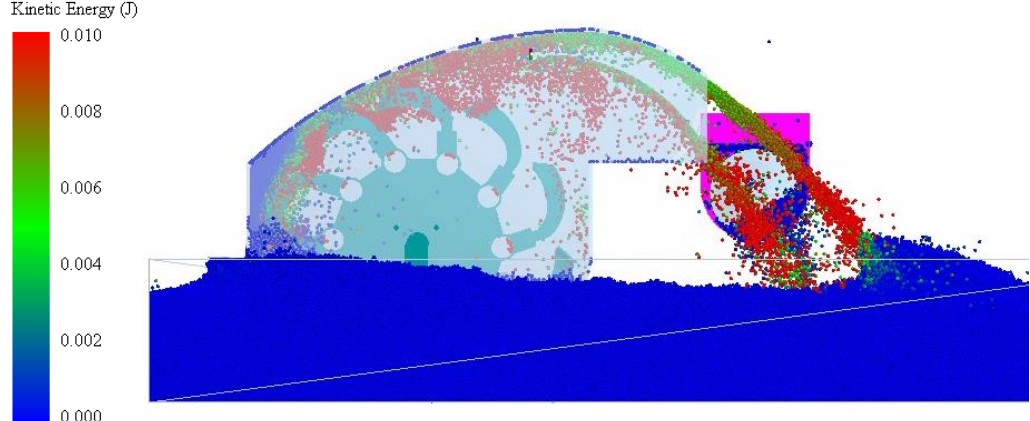

**Figure 7.** Soil fertilizer collision mixing mulch kinetic energy map.

In order to observe the flow direction of soil particles, the shape of soil particles is expressed as an arrow shape, and the direction of the arrow indicates the direction of particle movement. The operating characteristics of the machine at a travel speed of 0.3 m/s and a cutter speed of 170 r/min were used as an example.

As shown in Figure 8, when the soil groove was generated, the rotating tillage knife was not in contact with the soil particles, and the soil particles were not subject to the crushing and squeezing effect of the ditching knife on the soil. At this time, the speed of the soil particles was 0; there was no displacement trend. With the compound movement of the ditching knife driven by the knife shaft, the blade-side cutting edge began to contact

the soil, and the blade mainly through shear extrusion cut down the soil. At this time, the soil particles were thrown up by the ditching knife. With the ditching knife cutting soil volume increasing, the strength of the soil crushing increased in the ditching knife pushing action. The cut down soil has a tendency along the upward movement, and the soil kinetic energy gradually increases [28].

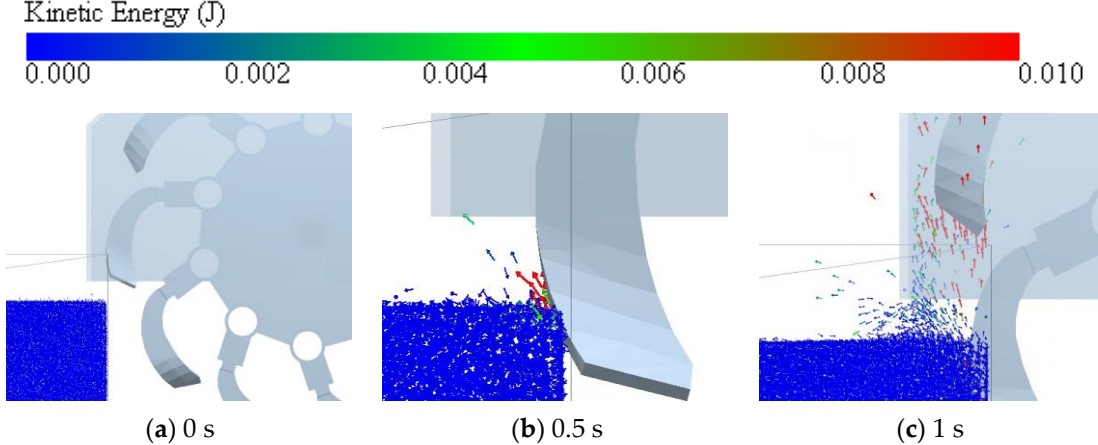

(**a**) 0 s  (**b**) 0.5 s  (**c**) 1 s

**Figure 8.** Soil cutting process. (**a**) the rotating tillage knife was not in contact with the soil. (**b**) the blade-side cutting edge began to contact the soil. (**c**) The contact area between the ditching knife cutter and the soil becomes larger.

As shown in Figure 9, the spreading of soil in the ditching process is often chaotic and irregular, but most of the soil particles will have the same trend and fall within a certain spreading range [29]. The deflector cover can deflect the soil in a predetermined direction when the cutter is spreading soil. The cut soil is thrown backward along the inflow surface under the action of the high-speed rotating ditching knife disk axis. When the kinetic energy is increasing, the particles are thrown to the splitting plate, and they are no longer subject to the action of the knife disk. Due to the inertia and internal force, the soil continues to maintain the movement. Under the action of the inflow plate, the backward thrown soil is divided into mixed fertilizer soil and mulch soil.

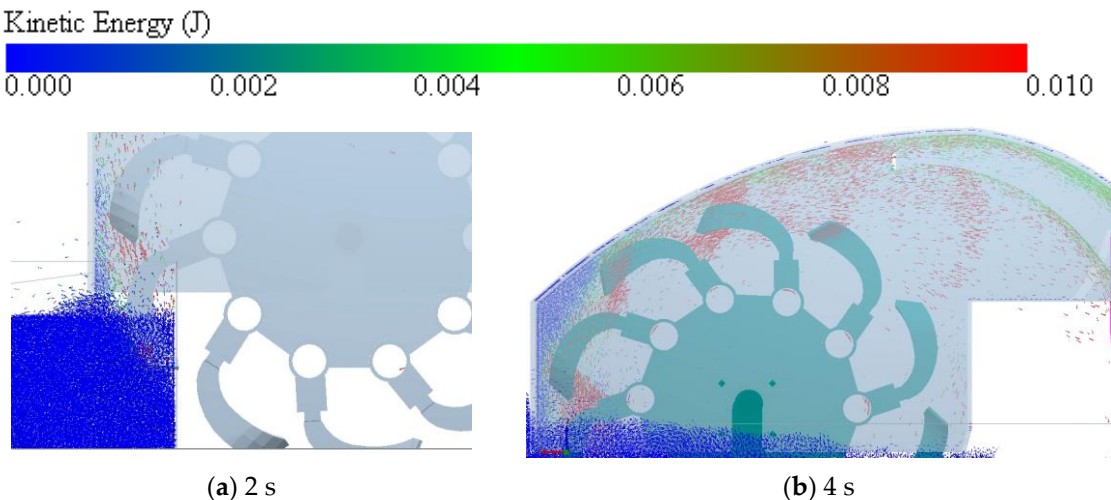

(**a**) 2 s  (**b**) 4 s

**Figure 9.** Soil throwing infusion process. (**a**) The soil is cut and thrown into the guide cover. (**b**) Trajectory of soil movement in the guide cover.

When the soil collides with the manure, the collision time is short, so the time of the collision process and the displacement occurred can be neglected, and only the kinetic energy change before and after the collision was considered. As shown in Figure 10,

the fertilizer soil mixture collides with the manure discharged by the auger. The kinetic energy is instantaneously reduced, and the collision changes the trajectory of movement for some soil particles, thus changing the landing point of the soil–fertilizer mixture. The soil–fertilizer mixture falls exactly to the ditch opened by the ditching device, and the soil mulched with fertilizer continues to do a backward throwing movement downward, covering the soil–fertilizer mixture, thus completing the ditching, fertilizer mixing and mulching operation.

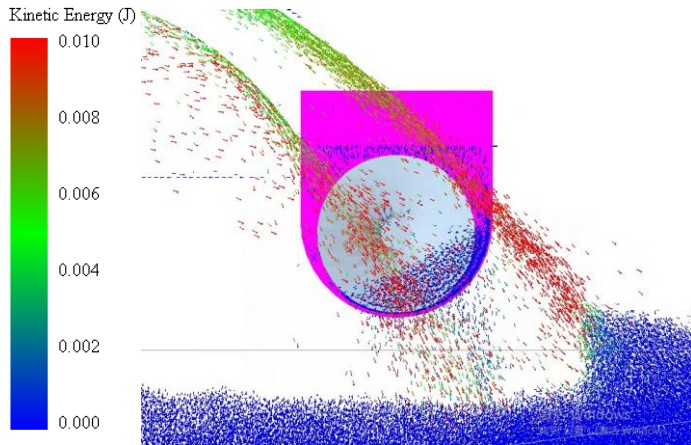

**Figure 10.** Soil fertilizer collision mixing mulching process.

### 2.3.3. Response Surface Experimental Design

Combined with the above analysis, the Box–Behnken test method was used. The test factors were blade speed, stirrer speed, and soil separating plate deflection angle. The factor levels were set based on the pre-experimental results. Three levels were set for each factor, and the test factors and levels are shown in Table 3. The soil–fertilizer collision mixing mulching device was designed to finally improve the fertilizer utilization rate, promote the effective absorption of fruit tree roots and solve the poor mulching quality, so the test indexes were mulching uniformity and soil–fertilizer mixing uniformity [30].

**Table 3.** Table of factor levels of the interaction test.

| Code | Turret Speed (r/min) | Stirrer Speed (r/min) | Soil Separator Deflection Angle (°) |
|------|------|------|------|
| −1 | 140 | 130 | 20 |
| 0 | 150 | 140 | 30 |
| 1 | 160 | 150 | 40 |

In the study of particle mixing, multiple samples are often collected, and the number of target particles in each sample is counted. Then, the coefficient of variation is calculated to characterize the mixing uniformity: the lower the coefficient of variation, the higher the mixing uniformity [31]. In this paper, we use the sampling method to calculate the soil fertilizer mixing uniformity. Firstly, we divide the particles in the fertilizer application trench into several samples and count the amount of manure in each sample. Then, we eliminate the samples with the total number of particles less than 20. Finally, we take the average of the results 10 times and calculate the homogeneity of soil–fertilizer mixing. The size of the samples was set as 60 mm × 60 mm × 60 mm cubes, with 20 samples in total.

Mixing uniformity is calculated with reference to the standard "NY/T 1024-2006 Technical Specification for Quality Evaluation of Feed Mixers" for the mixing uniformity M of stable manure and granular fertilizer.

$$M = \left(1 - \frac{S}{\overline{X}}\right) \times 100\% \tag{22}$$

$$S = \sqrt{\frac{\sum\limits_{i=1}^{n} \left(Xi - \overline{X}\right)^2}{n-1}} \quad (23)$$

where $S$—sample standard deviation.

$\overline{X}$—sample $X_1$, $X_2$ for each measurement . . . mean of $X_n$, kg.

$X_i$—mass of tracer in the sample, kg.

$n$—number of samples.

## 3. Results

### 3.1. Response Surface Experimental Results Analysis

A Box–Behnken response surface test was carried out, and the test scheme and results are shown in Table 4.

**Table 4.** Response surface test scheme and results.

| Test Number | Turret Speed $X_1$/(r/min) | Stirrer Speed $X_2$/(r/min) | Soil Separator Deflection Angle $X_3$/(°) | Mulch Uniformity $Y_1$/(%) | Soil and Fertilizer Mixing Uniformity $Y_2$/(%) |
|---|---|---|---|---|---|
| 1 | −1 | −1 | 0 | 90.68 | 88.44 |
| 2 | 1 | −1 | 0 | 94.84 | 81.89 |
| 3 | −1 | 1 | 0 | 95.24 | 85.23 |
| 4 | 1 | 1 | 0 | 91.47 | 87.14 |
| 5 | −1 | 0 | −1 | 96.86 | 89.77 |
| 6 | 1 | 0 | −1 | 94.85 | 84.25 |
| 7 | −1 | 0 | 1 | 96.31 | 86.12 |
| 8 | 1 | 0 | 1 | 96.38 | 75.13 |
| 9 | 0 | −1 | −1 | 90.21 | 80.38 |
| 10 | 0 | 1 | −1 | 93.37 | 86.44 |
| 11 | 0 | −1 | 1 | 96.00 | 83.12 |
| 12 | 0 | 1 | 1 | 94.15 | 76.97 |
| 13 | 0 | 0 | 0 | 90.66 | 87.48 |
| 14 | 0 | 0 | 0 | 92.46 | 90.28 |
| 15 | 0 | 0 | 0 | 92.44 | 90.26 |
| 16 | 0 | 0 | 0 | 91.55 | 89.27 |
| 17 | 0 | 0 | 0 | 91.66 | 87.68 |

ANOVA and significance ANOVA and significance tests were performed on the quadratic regression model, and the results are shown in Table 5.

**Table 5.** Analysis of variance for the cover uniformity back model.

| Project | Squares | df | Mean Square Error | F | p | Significance |
|---|---|---|---|---|---|---|
| Model | 74.36 | 9 | 8.26 | 8.50 | 0.0050 | ** |
| $X_1$ | 0.3003 | 1 | 0.3003 | 0.3090 | 0.5956 | - |
| $X_2$ | 0.7813 | 1 | 0.7813 | 0.8039 | 0.3997 | - |
| $X_3$ | 7.13 | 1 | 7.13 | 7.33 | 0.0303 | * |
| $X_{1\times2}$ | 15.72 | 1 | 15.72 | 16.18 | 0.0050 | ** |
| $X_1X_3$ | 1.08 | 1 | 1.08 | 1.11 | 0.3265 | - |
| $X_2X_3$ | 6.28 | 1 | 6.28 | 6.46 | 0.0386 | * |
| $X_1^2$ | 16.60 | 1 | 16.60 | 17.08 | 0.0044 | ** |
| $X_2^2$ | 1.96 | 1 | 1.96 | 2.02 | 0.1987 | - |
| $X_3^2$ | 23.46 | 1 | 23.46 | 24.14 | 0.0017 | ** |
| Residual | 6.80 | 7 | 0.9718 | | | |
| Lack of Fit | 4.59 | 3 | 1.53 | 2.76 | 0.1759 | - |
| Pure Error | 2.22 | 4 | 0.5541 | | | |
| Cor Total | 81.16 | 16 | | | | |

Note: $p \leq 0.01$ is highly significant, marked as **; $0.01 < p \leq 0.05$ is significant, marked as *; $p > 0.05$ is not significant, marked as -.

From the analysis of variance in Table 5, it can be seen that the regression model is extremely significant while loss of fit is not significant, indicating that the fitted regression equation of the model constructed for mulch uniformity is consistent with reality, and it can provide a better prediction of the relationship between mulch uniformity and each of the tested factors [32]. From the *p*-values in the table, it can be seen that the interaction of cutter speed and auger speed, the square of the cutter speed, and the square of the splitting plate deflection angle have a highly significant effect on the uniformity of mulching; the splitting plate deflection angle as well as the interaction of the auger speed and splitting plate deflection angle have a significant effect on the uniformity of mulching; and the interaction of cutter speed and the splitting plate deflection angle as well as the square of the auger speed have no significant effect on the mulching uniformity. Combined with the regression equation of uniformity of mulching, the analysis shows that the factors affecting the uniformity of mulching are mainly the square of the splitting plate deflection angle term $X_3{}^2$. Therefore, based on the ANOVA of each factor on the mulching uniformity, after excluding the non-significant, the relationship was

$$Y_1 = 91.75 + 0.94X_3 - 1.98X_1X_2 - 1.25X_2X_3 + 1.99X_1{}^2 - 2.36X_3{}^2 \tag{24}$$

Design-Expert 8.0.6 software was used to analyze the influences of turret speed, stirrer speed, and soil dividing plate deflection angle on mulching uniformity and mixing uniformity as well as the relationships among various factors, and to draw response surfaces, as shown in Figures 11 and 12.

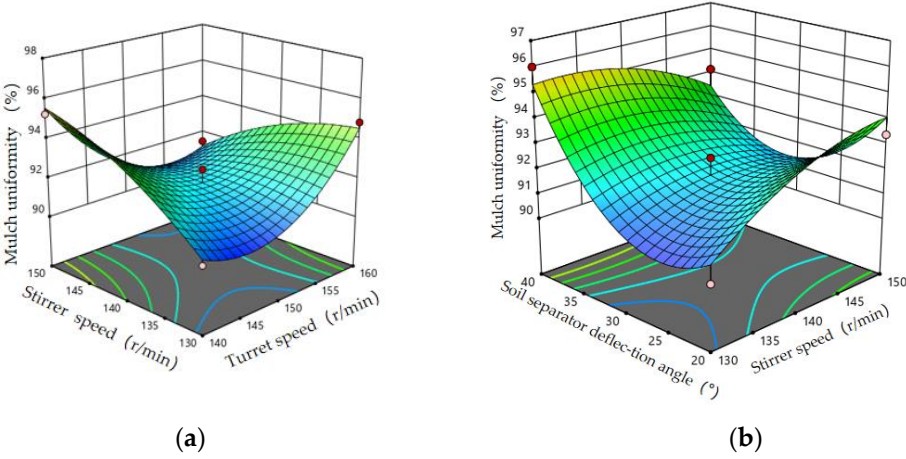

|        (a)        |        (b)        |

**Figure 11.** Effect of interaction factor on the mulching uniformity. (**a**) Influence of the interaction terms between turret speed and stirrer speed on the mulching uniformity. (**b**) Influence of the interaction terms of Stirrer speed and soil separator deflection angle on the mulching uniformity.

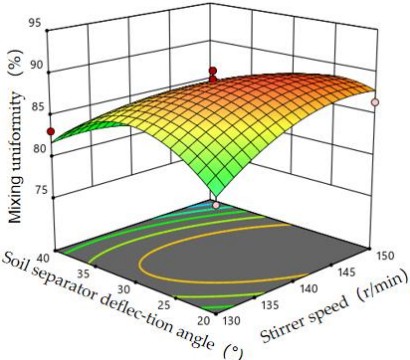

**Figure 12.** Influence of the interaction terms of Stirrer speed and soil separator deflection angle on the mixing uniformity.

Figure 11a shows that the mulching uniformity increases with the increase in the auger speed when the deflection angle of the plate is at the zero level, and the maximum value is achieved at 150 r/min of the auger speed; the mulching uniformity increases with the increase in the auger speed during the change of the blade speed from 130 to 160 r/min; the back-throwing speed of the soil increases during the increase in the blade speed, so that it quickly follows the guide hood located on the plate. As the rotating speed of the cutter increases, the back-throwing speed of the soil increases so that it quickly follows the deflector hood located on the splitting plate to carry out the mulching operation, which improves the mulching uniformity. The trend diagram shows that the interaction between the blade speed and the auger speed has a more significant effect. Figure 11b shows that the mulching uniformity decreases and then increases as the deflection angle of the splitting plate increases at zero level; the larger the deflection angle of the splitting plate, the more soil is mulched and the less soil is mixed, which will reduce the soil–fertilizer mixing uniformity. With the increase in auger speed, the uniformity of mulching soil increases, and the trend of change shows that the interaction between the deflection angle of the soil divider and the auger speed has a significant effect.

From the analysis of variance in Table 6, it can be seen that the regression model is extremely significant, and the loss of fit is not significant, which indicates that the fitted regression equation of the model constructed for mixing uniformity is consistent with the reality and can make a better prediction of the relationship between mixing uniformity and each test factor. From the $p$-values in the table, it can be seen that the influence of the deflection angle of the soil divider and the square of it on the mixing uniformity is highly significant. The square of the deflection angle of the soil divider and the square of the stirrer rotational speed have a significant influence on the mixing uniformity. Lastly, the interaction of the stirrer rotational speed and the deflection angle as well as the square of the stirrer rotational speed do not have a significant influence on the mixing uniformity. Therefore, according to the analysis of variance of each factor on the mixing uniformity, after excluding the insignificant, the relationship is as follows

$$Y_2 = 88.99 - 2.37A - 2.81C - 3.05BC - 2.33B^2 - 4.94C^2 \tag{25}$$

**Table 6.** Regression model analysis of variance for soil fertilizer mixing uniformity.

| Project | Squares | df | Mean Square Error | F | $p$ | Significance |
|---------|---------|----|----|-----|------|-------------|
| Model | 300.78 | 9 | 33.42 | 8.36 | 0.0053 | ** |
| $X_1$ | 41.18 | 1 | 41.18 | 10.31 | 0.0149 | * |
| $X_2$ | 0.4753 | 1 | 0.4753 | 0.119 | 0.7403 | - |
| $X_3$ | 63.28 | 1 | 63.28 | 15.84 | 0.0053 | ** |
| $X_1X_2$ | 17.89 | 1 | 17.89 | 4.48 | 0.0721 | - |
| $X_1X_3$ | 1.53 | 1 | 1.53 | 0.3817 | 0.5562 | - |
| $X_2X_3$ | 37.27 | 1 | 37.27 | 9.33 | 0.0185 | * |
| $X_1^2$ | 4.12 | 1 | 4.12 | 1.03 | 0.3436 | - |
| $X_2^2$ | 22.85 | 1 | 22.85 | 5.72 | 0.0481 | * |
| $X_3^2$ | 102.63 | 1 | 102.63 | 25.69 | 0.0015 | ** |
| Residual | 27.97 | 7 | 4 | | | |
| Lack of Fit | 20.62 | 3 | 6.87 | 3.74 | 0.1176 | - |
| Pure Error | 7.35 | 4 | 1.84 | | | |
| Cor Total | 328.75 | 16 | | | | |

Note: $p \leq 0.01$ is highly significant, marked as **; $0.01 < p \leq 0.05$ is significant, marked as *; $p > 0.05$ is not significant, marked as -.

As shown in Figure 12, the uniformity of fertilizer mixing increases from low level to high level with the deflection angle of the soil divider when the rotating speed of the cutter is at the zero level; the uniformity of fertilizer mixing increases with the increase in rotating speed during the change of rotating speed of the stirrer shaft from 130 to 150 r/min. The back-throwing speed of the soil increases during the process of the knife plate rotational

speed becoming larger, which makes the collision mixing effect more significant. From the change trend graph, it can be seen that the interaction effect of the soil dividing plate deflection angle and auger speed is more significant.

### 3.2. Optimization of Fertilizer Mixing Unit

On the basis of ensuring that each working parameter of the soil collision mixing and mulching device meets the agronomic requirements and the requirements of fertilization operations, optimization tests were conducted to obtain the optimal range of working parameters of it. To seek the optimal combination of working parameters, the Optimization module in Design-Expert 13 software was used to take the function of multi-objective optimization, with fertilizer mixing and mulching uniformity as the objective functions, and the blade speed, stirrer shaft speed, and splitting plate deflection angle as variables for optimization [33].

The mulch uniformity $Y_1$ and fertilizer mixing uniformity $Y_2$ are used as the working performance indexes of soil–fertilizer mixing and the fertilizer discharge device. Among the above response indexes, the mulch uniformity and fertilizer mixing uniformity should be improved as much as possible under the condition of meeting the requirements of fertilization operation. The optimized solution for the maximum fertilizer mixing uniformity is shown in Table 7.

$$\begin{cases} Max\{Y_1, Y_2\} \\ 140 \leq X_1 \leq 160 \\ 130 \leq X_2 \leq 150 \\ 20 \leq X_3 \leq 40 \end{cases} \tag{26}$$

**Table 7.** Optimal parameter combination scheme.

| | Turret Speed/(r/min) | Stirrer Shaft Speed/(r/min) | Soil Separator Deflection Angle/(°) | Mulch Uniformit/(%) | Soil and Fertilizer Mixing Uniformit/(%) |
|---|---|---|---|---|---|
| Numerical value | 140 | 146 | 22 | 96.86 | 88.35 |

### 3.3. Field Trials

To verify the effectiveness of the soil–fertilizer collision mixing mulching device, a field trial was conducted on 22 June 2022 at a small plant of the School of Mechanical and Electrical Engineering, Shihezi University. The manure used was produced by Xinjiang Shihezi Xinye Agriculture and Animal Husbandry Biological Development Co. Ltd., (Shihezi, China) and a John Deere 454 tractor was used to tow the soil–fertilizer collision mixing mulching test bed. The measurement tools used in the test were a soil moisture meter (TDR150), tape measure (range: 30 m, accuracy: 2 mm), steel ruler (range: 500 mm, index value: 1 mm), and inverter (AMB100-7R5G-T3 produced by Shenzhen Huichuan Technology Co. LTD, Foochow, China). The test site diagram is shown in Figure 13.

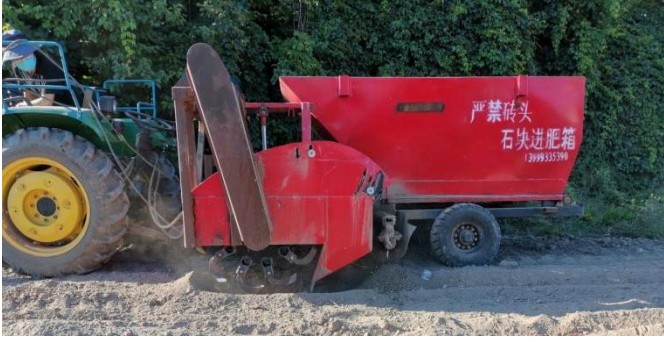

**Figure 13.** Soil and fertilizer mixing whole machine test.

In this test, because the physical properties of soil are similar to manure (both are in powder form), it is not possible to accurately calculate the mixing uniformity after mixing. Referring to the standard "NY/T 1024-2006 Technical Specification for Quality Evaluation of Feed Mixers", corn was used as tracer particles; before the soil–fertilizer mixing test, corn has been mixed with manure. A flat and smooth concrete floor was selected; the length of the measuring area is 10 m, and the width covers one working width of the machine. The device stably goes through the measuring area to complete the fertilizer discharge operation. After the test, 50 cm front and rear of the fertilizer in the test area were removed, and the remaining part was divided into not less than 30 consecutive equal sections according to the length of 10 cm. The mass of corn in each small section was collected and measured to evaluate the uniformity of mixing of corn as tracer particles in the manure. The uniformity of corn mixing in the manure was calculated to be 98.53%.

As there was soil flow during the sampling and measurement process, the actual value could not be measured accurately. So, to ensure that the soil–fertilizer mixture results were closer to the actual value, two acrylic plates were taken and inserted into the bottom of the trench and fixed before sampling, as shown in Figure 14. The two plates were spaced 10 cm apart, and the trench on one side of the plates was cleaned up so that the soil–fertilizer mixture and the overall condition of the mulch could be seen through the transparent plates. The mulching soil between the plates was first taken out with a small shovel and weighed for the mass so as to find out the uniformity of that. Afterwards, the soil–fertilizer mixture in the middle was removed and sieved to find the mass of corn. Each group test was repeated five times to find the mean and standard deviation of the five times for the calculation of soil–fertilizer mixture uniformity [34,35].

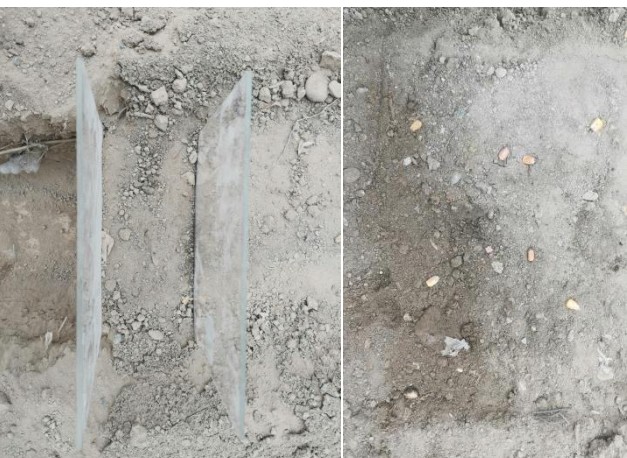

**Figure 14.** Sampling process.

The soil–fertilizer mixing uniformity is sensitive to sample size. In order to verify the accuracy of the discrete element model and the reliability of the simulation optimization results, the sample size is set to be consistent with the bench test in the simulation test with the optimal parameters. The test results of soil cover uniformity and soil and fertilizer mixture uniformity are shown in Table 8, and the soil cover uniformity is 94.37%, while the mixing uniformity was 87.02%. The results show that the discrete element model is accurate, the simulation optimization results are reliable, and the mixing performance of the soil–fertilizer collision mixing mulching device processed according to the optimized parameters is good.

**Table 8.** Optimization results test values.

| Number of Tests | Soil Quality of Fertilizer cover/g | Mass of Soil–Fertilizer Mixture/g | Tracer Particle Mass/g | Mulch Uniformity/% | Soil and Fertilizer Mixing Uniformity/% |
|---|---|---|---|---|---|
| 1 | 1622.5 | 5800 | 23 | | |
| 2 | 1703.5 | 6061 | 19 | | |
| 3 | 1554 | 5558 | 18.5 | | |
| 4 | 1616.5 | 5473 | 24.5 | | |
| 5 | 1751.5 | 5864 | 19 | 94.37 | 87.02 |
| 6 | 1538 | 5751 | 19.6 | | |
| 7 | 1776 | 6647 | 21.5 | | |
| 8 | 1797 | 5448 | 20 | | |
| 9 | 1584.5 | 5808 | 20.3 | | |
| 10 | 1631 | 5429 | 23 | | |

## 4. Discussion

Firstly, the apple-growing areas were researched, and relevant information was collected and organized. Insisting on the principle of combining agricultural machinery and agronomy, a technical scheme of soil–fertilizer mixing and mulching device for deep fertilizer application machine was proposed. Secondly, the conveying of manure particles in the spiral auger discharge parts and of soil in the ditching knife and guide cover were analyzed. Thirdly, the key components of the device were designed, and a simulation test was conducted to analyze the mixing process using EDEM. Finally, the working parameters of the device were optimized according to the test results.

## 5. Conclusions

(1) The design of the soil–fertilizer mixing and mulching device for the deep application of manure was carried out, and the key parameters of the ditching mechanism, soil-dividing plate and auger dragon were determined, which can provide a reference for the research of soil–fertilizer mixing and application technology in a Xinjiang apple orchard.

(2) The simulation test of soil and fertilizer mixing and mulching process was conducted by EDEM. When the rotational speed of the cutter, stirrer and deflection angle of the dividing plate were 140 r/min, 146 r/min and 22°, respectively, the highest uniformity of mulching was 96.86%, and the largest uniformity of mixing was 88.35%.

(3) In the soil–fertilizer mixing and mulching bench test, the test results show that the average fertilizer mixing uniformity is 87.02%, with an average error of 1.33% from the optimal value of the simulation test, and the uniformity of mulching is 94.37%, with an average error of 2.49%, which meets the agronomic requirements. It shows that the discrete element model is accurate, the simulation optimization results are reliable, and the soil–fertilizer collision mixing mulching device processed according to the optimized parameters has better mixing performance and can meet the operation requirements.

**Author Contributions:** Conceptualization, Y.N. and H.M.; methodology, J.Q.; software, Y.N. and J.Z.; experiment, Y.N. and J.L.; data curation, Y.N. and J.Z.; writing—original draft preparation, Y.N.; writing—review and editing, Y.N. and J.Q.; visualization, H.P.; supervision, H.P.; funding acquisition, H.M. and J.Q. All authors have read and agreed to the published version of the manuscript.

**Funding:** This research was funded by Corps Science and Technology Plan Projects, grant number "2022CB002-03"; Shihezi University Youth Innovation Incubation Talent Program Project, grant number "CXPY202118"; Autonomous Region Key R&D Task Special Projects, grant number "2022B02028-2".

**Institutional Review Board Statement:** Not applicable.

**Data Availability Statement:** The data presented in this study are available on request from the corresponding author.

**Acknowledgments:** The authors would like to thank the teachers and supervisors for their technical support. We would also like to acknowledge the assistance provided by brothers and sisters during the tests. Finally, we are grateful to the editor and anonymous reviewers for providing helpful suggestions to improve the quality of this paper.

**Conflicts of Interest:** The authors declare no conflict of interest.

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
