# Peer review of "Design and Test of Soil–Fertilizer Collision Mixing and Mulching Device for Manure Deep Application Machine"

_agriculture, doi:10.3390/agriculture13030709_

Round 1

Reviewer 1 Report

The excellent article

Design and test of soil-fertilizer collision mixing and mulching device for orchard stable fertilizer deep application machine

Was reviewed and the following considerations obtained:

Also I would like to ask what is the effect of this machine operation in the apple orchard as it cuts the roots. Fruit trees have 1 mm roots used to suck the water from soil and sometimes we add fertilizers with care to avoid root damage. We use fertigation to avoid root damage.

Considerations:

In the abstract line 17, eliminate all the words after respectively.

In line 15 add maximum, The simulation results showed that the maximum mixing

In line 65 eliminate “and mixed”.

Eliminate in line 82, “, and conducting field trials for verification,”. Add a full stop after device.

In line 84 where the paragraph ends, add a comment that field trials were conducted for verification.

In line 127 after the parenthesis eliminate the point and put a comma before the words as shown…

Why equation 6 has a power of 2.5?

In line 191 remove the comma and insert a point after effect. Then if will have a capital letter. In line 192 after circumference add “and”. In line 193 after blade put a stop and eliminate the comma.

What is “conducive” in line 198?

In line 199 reference cannot be 30 as it goes in order. If it is 30 all the references after 20 have to be moved.

In line 209 what does EDEM mean?

In equation of line 217 what is the meaning of G and N?

From reference 27 in line 245 you jump to reference 31 in line 305. Where are references 28, 29 and 30.

The sentence in line 309 has to be rewritten.

Please rewrite line 320 as it is not readable.

In line 329 eliminate “is” at the end of the line

In fig 11 can you add the churn speed and blade speed in the axis?

The word “is” in line 364 should be changed.

Eliminate in line 393 the first uniformity word.

The sentence from line 443 to 447 should be sliced so thet it can be more readable.

Eliminate line 472 and move largest to the end of line 471, so that text ends as “the largest uniformity of mulching” and in line 472 include highest in “96.86% and the highest uniformity”

Please rewrite section 3 of the conclusions as it is not clear and sentences are too long. If you make the sentences shorter it will be clearer.

References:

References 2, 4 and 33 have capital letters in the authors and should be fixed.

All the references after the article title have [J] which should be removed.

Publication year in articles must be in bold letter. Between number of journal and pages there should be a comma and not a semi colon.

Reviewer 2 Report

Abstract have around 223 words

At line

48 you have an extra "." or you have missing space between "mixing" and "M".Also why don't use the same order of names like in reference list "Kwapinska M."

51  Insert extra space between "number."

57  You say "Huang Yue et al" but at reference list you have only one author "Huang Y." the

If you use autocorrect you will see all missing space between words, "," and "."

In my opinion at line 95-97 you must correct "device.1. lifting frame 2. guide cover 3. trenching mechanism 4. trench cleaning plate 5. soil dividing plate 6. churn 7. fer-tilizer box 8. frame." to "device. Where: 1. lifting frame; 2. guide cover; 3. trenching mechanism; 4. trench cleaning plate; 5. soil dividing plate; 6. churn; 7. fer-tilizer box; 8. frame."

Try to corect all the same text in figures explication.

In my pdf line 117 "Table 1. Main technical parameters of the whole machine." are the last line in page, try to move this line in next page  //// same at line 212 "2.2.3. Soil separation plate" //// same at line 322 "Table 4. Response surface test scheme and results."

212 Missing in description of Figure 6 "a" and "b"

296 Correct "results. 3 levels" to "results. Three levels"

302 Missing space between table and text

328-329 Missing space between explication text and paper text, also explication text must with smaller font size (9) /// The same observation for 362-364

342 Where is "superscript 2" at "X" or at "3" ?

343 Missing some explication for Figure 11 before figure.

376 Missing some explication for Figure 12 before figure.

379 Correct "Fig 12" to "Figure 12"

438 Ad space between text and figure

462 Ad some explication text for conclusions.

514 you have an extra space between "E" and "."

518 between names missing space insert the necesary space afetr "," and after "." at "Xiao H,Zhao Y,Ding W,Mei S,Han Y,Zhang Y,Yan H,Song Z.Design"

You must correct all reference list for missing space between words, comma "," and dot "."

For example at line 501 "Technology,2019(11):48." correct to "Technology, 2019 (11):48." /// ay line 507 "organic fertilizers[J]. Modern Agricultural Science and Technology,2020(12):199-200." to "organic fertilizers [J]. Modern Agricultural Science and Technology, 2020 (12):199-200."
